# VISUAL IMITATION WITH REINFORCEMENT LEARNING USING RECURRENT SIAMESE NETWORKS

## ABSTRACT

It would be desirable for a reinforcement learning (RL) based agent to learn behaviour by merely watching a demonstration. However, defining rewards that facilitate this goal within the RL paradigm remains a challenge. Here we address this problem with Siamese networks, trained to compute distances between observed behaviours and the agent's behaviours. Given a desired motion such Siamese networks can be used to provide a reward signal to an RL agent via the distance between the desired motion and the agent's motion. We experiment with an RNN-based comparator model that can compute distances in space and time between motion clips while training an RL policy to minimize this distance. Through experimentation, we have had also found that the inclusion of multi-task data and an additional image encoding loss helps enforce the temporal consistency. These two components appear to balance reward for matching a specific instance of a behaviour versus that behaviour in general. Furthermore, we focus here on a particularly challenging form of this problem where only a single demonstration is provided for a given task – the one-shot learning setting. We demonstrate our approach on humanoid agents in both 2D with 10 degrees of freedom (DoF) and 3D with 38 DoF.

## 1 INTRODUCTION

Imitation learning and Reinforcement Learning (RL) often intersect when the goal is to imitate with incomplete information, for example, when imitating from motion capture data (mocap) or video. In this case, the agent needs to search for actions that will result in observations similar to the expert. However, formulating a metric that will provide a reasonable distance between the agent and the expert is difficult. Robots and people plan using types of internal and abstract pose representations that can have reasonable distances; however, typically when animals observe others performing tasks, only visual information is available. Using distances in pose-space is ill-suited for imitation as changing some features can result in drastically different visual appearance. In order to understand how to perform tasks from visual observation a mapping/transformation is used which allows for the minimization of distance in appearance. Even with a method to transform observations to a similar pose space, each person has different capabilities. Because of this, people are motivated to learn transformations in space and time where they can reproduce the behaviour to the best of their own ability. How can we learn a representation similar to this latent space?

An essential detail of imitating demonstrations is their sequential and causal nature. There is both an ordering and speed in which a demonstration is performed. Most methods require the agent to learn to imitate the temporal and spatial structure at the same time creating a potentially narrow solution space. When the agent becomes desynchronized with the demonstration, the agent will receive a low reward. Consider the case when a robot has learned to stand when its goal is to walk. Standing is spatially close to the demonstration and actions that help the robot stand, as opposed to falling, should be encouraged. How can such latent goals be encouraged?

If we consider a phase-based reward function $r = R(s, a, \phi)$ where $\phi$ indexes the time in the demonstration and $s$ and $a$ is the agent state and action. As the demonstration timing $\phi$, often controlled by the environment, and agent diverge, the agent receives less reward, even if it is visiting states that exist elsewhere in the demonstration. The issue of determining if an agent is displaying out-of-phase behaviour can understood as trying to find the $\phi$ that would result in the highest reward

$\phi' = \max_\phi R(s, a, \phi)$ and the distance $\phi' - \phi$ is an indicator of how far away in *time* or out-of-phase the agent is. This phase-independent form can be seen as a form of reward shaping. However, this naive description ignores the ordered property of demonstrations. What is needed is a metric that gives reward for behaviour that is in the proper order, independent of phase. This ordering motivates the creation of a recurrent distance metric that is designed to understand the context between two motions. For example, does this motion look like a walk, not, does this motion look precisely like that walk.

Our proposed Visual Imitation with Reinforcement Learning (VIRL) method uses Recurrent Siamese Networks (RSNs) and has similarities to both Inverse Reinforcement Learning (IRL) (Abbeel & Ng, 2004) and Generative Advisarial Imitation Learning (GAIL) (Ho & Ermon, 2016). The process of learning a cost function that understands the space of policies to find an optimal policy given a demonstration is fundamentally IRL. While using positive examples from the expert and negative examples from the policy is similar to the method GAIL uses to train a discriminator to recognize in distribution examples. In this work, we build upon these techniques by constructing a method that can learn policies using noisy visual data without action information. Considering the problem's data sparsity, we include data from other tasks to learn a more robust distance function in the space of visual sequence. We also construct a cost function that takes into account the demonstration ordering as well as pose using a recurrent Siamese network. Our contribution consists of proposing and exploring these forms of recurrent Siamese networks as a way to address a critical problem in defining reward structure for imitation learning from the video for deep RL agents and accomplishing this on simulated humanoid robots for the challenging single shot learning setting.

## 2 RELATED WORK

**Learning From Demonstration** Searching for good distance functions is an active research area (Abbeel & Ng, 2004; Argall et al., 2009). Given some vector of features, the goal is to find an optimal transformation of these features, such in this transformed space, there exists a strong contextual meaning. Previous work has explored the area of state-based distance functions, but most rely on pose based metrics (Ho & Ermon, 2016; Merel et al., 2017) that come from an expert. While there is other work using distance functions, including for example Sermanet et al. (2017); Finn et al. (2017); Liu et al. (2017); Dwibedi et al. (2018), few use image based inputs and none consider the importance of learning a distance function in time as well as space. In this work, we train recurrent Siamese networks (Chopra et al., 2005) to learn distances between videos.

**Partially Observable Imitation Without Actions** For Learning from Demonstration (LfD) problems the goal is to replicate the behaviour of expert $\pi_E$ behaviour. Unlike the typical setting for humans learning to imitate, LfD often assumes the availability of expert action and observation data. Instead, in this work, we focus on the case where only noisy actionless observations of the expert are available. Recent work uses Behavioural Cloning (BC) to learn an inverse dynamics model to estimate the actions used via maximum-likelihood estimation (Torabi et al., 2018). Still, BC often needs many expert examples and tends to suffer from state distribution mismatch issues between the *expert* policy and *student* (Ross et al., 2011). Work in (Merel et al., 2017) proposes a system based on GAIL that can learn a policy from a partial observation of the demonstration. In this work, the discriminator's state input is a customized version of the expert's state and does not take into account the demonstration's sequential nature. The work in (Wang et al., 2017) provides a more robust GAIL framework along with a new model to encode motions for few-shot imitation. This model uses an Recurrent Neural Network (RNN) to encode a demonstration but uses expert state and action observations. In our work, the agent is limited to only a partial visual observation as a demonstration. Additional works learn implicit models of distance (Yu et al., 2018; Pathak et al., 2018; Finn et al., 2017; Sermanet et al., 2017), none of these explicitly learn a sequential model considering the demonstration timing. An additional version of GAIL, infoGAIL (Li et al., 2017), included pixel based inputs. Goals can be specified using the latent space from a Variational Auto Encoder (VAE) (Nair et al., 2018). Our work extends this VAE loss using sequence data to train a more temporally consistent latent representation. Recent work (Peng et al., 2018b) has a 2D control example of learning from video data. We show results on more complex 3D tasks and additionally model distance in time. In contrast, here we train a recurrent siamese model that can be used to en-

able curriculum learning and allow for computing distances even when the agent and demonstration are out of sync.

## 3 PRELIMINARIES

In this section, we outline the general RL framework and specific formulations for RL that we rely upon when developing our method in Section 4.

**Reinforcement Learning**    Using the RL framework formulated with a Markov Dynamic Process (MDP): at every time step $t$, the world (including the agent) exists in a state $s_t \in S$, wherein the agent is able to perform actions $a_t \in A$, sampled from a policy $\pi(a_t|s_t)$ which results in a new state $s_{t+1} \in S$ and reward $r_t$ according to the transition probability function $T(r_t, s_{t+1}|s_t, a_t)$. The policy is optimize to maximize the future discounted reward

$$J(\pi) = \mathbb{E}_{r_0,\ldots,r_T}\left[\sum_{t=0}^{T}\gamma^t r_t\right], \tag{1}$$

where $T$ is the max time horizon, and $\gamma$ is the discount factor, indicating the planning horizon length. Inverse reinforcement learning refers to the problem of extracting a reward function from observed optimal behavior Ng et al. (2000). In contrast, in our approach we learn a distance that works across a collection of behaviours. Further, we do not assume the example data to be optimal. See Appendix 7.2 for further discussion of the connections of our work to inverse reinforcement learning.

**GAIL**    VIRL is similar to the GAIL framework (Ho & Ermon, 2016) which uses a Generative Advasarial Network (GAN) (Goodfellow et al., 2014), where the discriminator is trained with positive examples from the expert trajectories and negative examples from the policy. The generator is a combination of the environment, policy and current state visitation probability induced by the policy $p_\pi(s)$.

$$\min_{\theta_\pi}\max_{\theta_\phi}\mathbb{E}_{\pi_E}[\log(D(s, a|\theta_\phi))] + \mathbb{E}_{\pi_{\theta_\pi}}[\log(1 - D(s, a|\theta_\phi))] \tag{2}$$

In this framework the discriminator provides rewards for the RL policy to optimize, as the probability of a state generated by the policy being in the distribution $r_t = D(s_t, a_t|\theta_\phi)$. While this framework has been shown to work in practice, this dual optimization is often unstable. In the next section we will outline our method for learning a more stable distance based reward over sequences of images.

## 4 CONCEPTUAL DISTANCE-BASED REINFORCEMENT LEARNING

Our approach is aimed at facilitating imitation learning within an underlying RL formulation over partially observed observations $o$. Unlike the situation in GAIL, we do not rely on having accces to state, $s$ and action, $a$ information – our idea is to minimize a function that determintes the distance between two sequences observations, $\mathbf{o}$, one from the desired example behavior $\mathbf{o}^e$, and another from the current agent behavior $\mathbf{o}^a$. We can then define the reward used within an underlying RL framework in terms of a distance function $D$, such that

$$r_{\hat{t}}(\mathbf{o}^e, \mathbf{o}^a) = -D(\mathbf{o}^e, \mathbf{o}^a, \hat{t}) = \sum_{t=0}^{\hat{t}} -d(o_t^e, o_t^a), \tag{3}$$

where in our setting here $D(\mathbf{o}^e, \mathbf{o}^a, \hat{t})$ models a distance between video clips from time $t = 0$ to $\hat{t}$.

A simple formulation of the approach above can be overly restrictive on sequence timing. While these distances can serve as RL rewards, they often provide insufficient signal for the policy to learn a good imitative behaviour, especially when the agent only has partial observations of the expert. We can see an example of this in Figure 1a were starting at $t_5$ the agent (in red) begins to exhibit behaviour that is similar to the expert (in blue) yet the spatial distance indicates that this state is further away from the desired behaviour than at $t_4$.

To encurage the agent to match any part of the expert behaviour we propose decomposing the distance into two distances, by adding a type of temporal distance shown in green. To compute a time

independant distance we can find the state in the expert sequence that is closest to the agent's current state $\arg \min_{\hat{t} \in T} d(\mathbf{o}_{\hat{t}}^e, o_t^a)$ and use it in the following distance measure

$$d^T(\mathbf{o}^e, \mathbf{o}^a, \hat{t}, t) = \ldots + d(\mathbf{o}_{\hat{t}-1}^e, \mathbf{o}_{t-1}^a) + d(\mathbf{o}_{\hat{t}}^e, \mathbf{o}_t^a) + d(\mathbf{o}_{\hat{t}+1}^e, \mathbf{o}_{t+1}^a) + \ldots \tag{4}$$

Using only a single state time-alined may lead to the agent fixating on matching a single state in the expert demonstration. To avoid this the neighbouring states given sequence timing readjustment are used in the distance computation. This framework allows the agent to be rewarded for exhibiting behaviour that matches any part of the experts demonstration. The better is learns to match parts of the expert demonstration the more reward it is given. The previous spatial distance will then help the agent learn to sync up its timing with the deomonstration. Next we describe how we learn both of these distances.

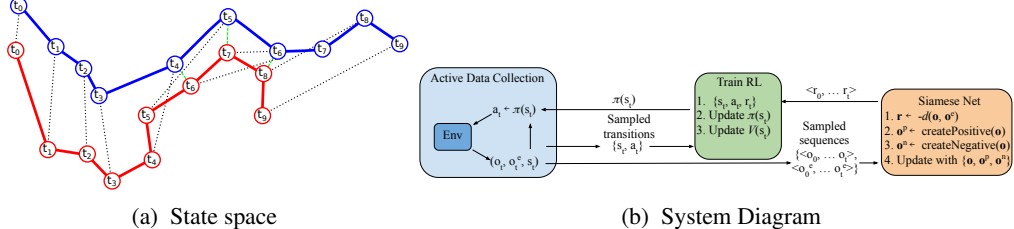

(a) State space            (b) System Diagram

Figure 1: A pair of sequences from the expert (blue) and agent (red). The spatial distance is shown via doted black lines and the temporal distance for $d(\mathbf{o}^e, o_7^a)$ is visulized in green. On the right the flow of control for the learning system is shown.

**Distance Metric Learning**   Many methods can be used to learn a distance function in state-space. Here we use a Siamese network $f(o^e, o^a)$ with a triplet loss over time and task data (Chopra et al., 2005). The triplet loss is used to minimize the distance between two examples that are *positive*, very similar or from the same class, and maximize the distance between pairs of examples that are known to be unrelated. For more details see supplementary document.

**Sequence Imitation**   The distance metric is formulated in a recurrent style where the distance is computed from the current state and conditioned on all previous states $d(o_t | o_{t-1}, \ldots, o_0)$. The loss function is a combination of distance Eq. 9 and VAE-based representation learning objectives from Eq. 7 and Eq. 8, detailed in the supplementary material. This combination of sequence-based losses assists in compressing the representation while ensuring intermediate representations are informative. The loss function used to train the distance model on a *positive* pair of sequences is:

$$\mathcal{L}_{VIRL}(\mathbf{o}_i, \mathbf{o}_p, \cdot) = \lambda_0 \mathcal{L}_{SN}(\mathbf{o}_i, \mathbf{o}_p, \cdot) + \lambda_1 \left[ \frac{1}{T} \sum_{t=0}^{T} \mathcal{L}_{SN}(o_{i,t}, o_{p,t}, \cdot) \right] +$$

$$\lambda_2 \left[ \frac{1}{T} \sum_{t=0}^{T} \mathcal{L}_{VAE}(o_{i,t}) + \mathcal{L}_{VAE}(o_{p,t}) \right] +$$

$$\lambda_3 \left[ \mathcal{L}_{AE}(\mathbf{o}_i) + \mathcal{L}_{AE}(\mathbf{o}_p) \right].$$

Where $\lambda = \{0.7, 0.1, 0.1, 0.1\}$. With a negative pair, the second sequence used in the VAE and AE losses would be the negative sequence.

The Siamese loss function remains the same as in Eq. 9 but the overall learning process evolves to use an RNN-based deep networks. A diagram of the full model is shown in Figure 2. This model uses a time distributed Long Short-Term Memory (LSTM). A single convolutional network $\texttt{conv}^a$ is first used to transform images of the demonstration $\mathbf{o}^a$ to an encoding vector $e_t^a$. After the sequence of images is distributed through $conv^a$ there is an encoded sequence $< e_0^a, \ldots, e_t^a >$, this sequence is fed into the RNN $lstm^a$ until a final encoding is produced $h_t^a$. This same process is performed for a copy of the RNN $lstm^a$ producing $h_t^b$ for the agent $\mathbf{o}^b$. The loss is computed in a similar fashion to (Mueller & Thyagarajan, 2016) using the sequence outputs of images from the agent and another from the demonstration. The reward at each timestep is computed as $r_t =$

$$||h_t^a - h_t^b|| + ||e_t^a - e_t^b|| = ||lstm^a(conv_a(s_t^a)) - lstm^a(conv_a(s_t^b))|| + ||conv_a(s_t^a) - conv_a(s_t^b)||.$$

At the beginning of each episode, the RNN's internal state is reset. The policy and value function have 2 hidden layers with 512 and 256 units, respectively. The use of additional VAE-based image and Auto Encoder (AE)-based sequence decoding losses improve the latent space conditioning and representation.

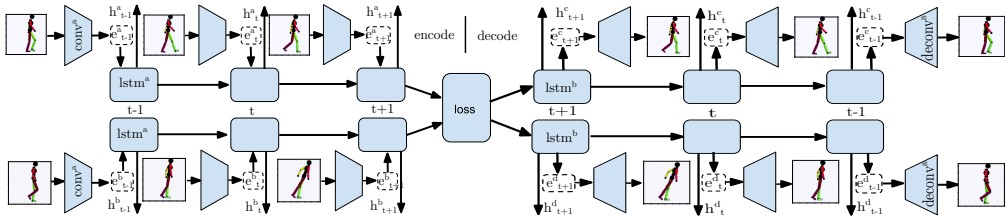

Figure 2: Siamese network structure. The convolutional portion of the network includes 2 convolution layers of 8 filters with size $6 \times 6$ and stride $2 \times 2$, 16 filters of size $4 \times 4$ and stride $2 \times 2$. The features are then flattened and followed by two dense layers of 256 and 64 units. The majority of the network uses ReLU activations except for the last layer that uses a sigmoid activation. Dropout is used between convolutional layers. The RNN-based model uses a LSTM layer with 128 hidden units, followed by a dense layer of 64 units. The decoder model has the same structure in reverse with deconvolution in place of convolutional layers.

**Unsupervised Data labelling** To construct *positive* and *negative* pairs for training we make use of time information in a similar fashion to (Sermanet et al., 2017), where observations at similar times in the same sequence are often correlated and observations at different times will likely have little similarity. We compute pairs by altering one sequence and comparing this modified version to its original. Positive pairs are created by adding noise to the sequence or altering a few frames of the sequences. Negative pairs are created by shuffling one sequence or reversing it. More details are available in the supplementary material. Imitation data for 24 other tasks are also used to help condition the distance metric learning process. These include motion clips for running, backflips, frontflips, dancing, punching, kicking and jumping along with the desired motion. For details on how positive and negative pairs are created from this data, see the supplementary document.

---

**Algorithm 1** Learning Algorithm

Initialize model parameters $\theta_\pi$ and $\theta_d$
Create experience memory $D \leftarrow \{\}$
**while** not done **do**
  **for** $i \in \{0, \dots N\}$ **do**
    $\tau_i \leftarrow \{\}$
    $\{s_t, o_t^e, o_t^a\} \leftarrow$ env.reset()
    **for** $t \in \{0, \dots, T\}$ **do**
      $a_t \leftarrow \pi(\cdot|s_t, \theta_\pi)$
      $\{s_{t+1}, o_{t+1}^e, o_{t+1}^a\} \leftarrow$ env.step($a_t$)
      $r_t \leftarrow -d(o_{t+1}^e, o_{t+1}^a|\theta_d)$
      $\tau_{i,t} \leftarrow \{s_t, o_t^e, o_t^a, a_t, r_t\}$
      $\{s_t, o_t^e, o_t^a\} \leftarrow \{s_{t+1}, o_{t+1}^e, o_{t+1}^a\}$
    **end for**
  **end for**
  $D \leftarrow D \bigcup \{\tau_0, \dots, \tau_N\}$
  Update $d(\cdot)$ parameters $\theta_d$ using $D$
  Update policy $\theta_\pi$ using $\{\tau_0, \dots, \tau_N\}$
**end while**

---

Importantly the RL environment generates two different state representations for the agent. The first state $s_{t+1}$ is the internal robot pose. The second state $o_{t+1}$ is the agent's rendered view, shown in Figure 2. The rendered view is used with the distance metric to compute the similarity between the agent and the demonstration. We attempted using the visual features as the state input for the policy as well; this resulted in poor policy quality. Details of the algorithm used to train the distance metric and policy are outlined in the supplementary document Algorithm 1.

## 5 ANALYSIS AND RESULTS

The simulation environment used in the experiments is similar to the DeepMind Control Suite (Tassa et al., 2018). In this simulated robotics environment, the agent is learning to imitate a given reference motion. The agent's goal is to learn a policy to actuate Proportional Derivative (PD) controllers at 30 fps to mimic the desired motion. The simulation environment provides a hard-coded reward function based on the robot's pose that is used to evaluate the policy quality. The demonstration $M$ the agent is learning to imitate is generated from a clip of mocap data. The mocap data is used to

animate a second robot in the simulation. Frames from the simulation are captured and used as video input to train the distance metric. The images captured from the simulation are converted to grey-scale with $64 \times 64$ pixels. We train the policy on pose data, as link distances and velocities relative to the robot's Centre of Mass (COM). This simulation environment is new and has been created to take motion capture data and produce multi-view video data that can be used for training RL agents or generating data for computer vision tasks. The environment includes challenging and dynamic tasks for humanoid robots. Some example tasks are imitating running, jumping, and walking, shown in Figure 3 and *humanoid2d* detailed in the supplementary material.

**3D Humanoid Robot Imitation**    In these simulated robotics environments the agent is learning to imitate a given reference motion of a walk, run, jump or zombie motion. A single motion demonstration is provided by the simulation environment as a cyclic motion. During learning, we include additional data from all other tasks for the walking task this would be: walking-dynamic-speed, running, jogging, frontflips, backflips, dancing, jumping, punching and kicking) that are only used to train the distance metric. We also include data from a modified version of the tasks that has a randomly generated speed modifier $\omega \in [0.5, 2.0]$ walking-dynamic-speed, that warps the demonstration timing. This additional data is used to provide a richer understanding of distances in space and time to the distance metric. The method is capable of learning policies that produce similar behaviour to the expert across a diverse set of tasks. We show example trajectories from the learned policies in Figure 3 and in the supplemental Video. It takes $5 - 7$ days to train each policy in these results on a $16$ core machine with an Nvidia GTX1080 GPU.

**Algorithm Analysis and Comparison**    To evaluate the learning capabilities and improvements of VIRL we compare against two other methods that learn a distance function in state space, GAIL and using a VAE to train an encoding and compute distances between those encodings, similar to (Nair et al., 2018), using the same method as the Siamese network in Figure 4a. We find that the VAE alone does not appear to capture the critical distances between states, possibly due to the decoding transformation complexity. Similarly, the GAIL baseline produces very jerky motion or stands still, both of which are contained in the imitation distribution. Our method that considers the temporal structure of the data learns faster and produces higher value policies.

Additionally, we create a multi-modal version of VIRL. Here we replace the bottom conv net with a dense network and learn a distance metric between agent poses and imitation video. The results of these models, along with the default manual reward function provided by the environment, are shown in Figure 4b. The multi-modal version appears to perform about equal to the vision-only modal. In Figure 4b we also compare our method to a non-sequence-based model that is equivalent to Time Contrastive Network (TCN). On average VIRL achieves higher value policies. We find that using the RNN-based distance metric makes the learning process more gradual. We show this learning effect in Figure 4b, where the original manually created reward with flat feedback leads to slow initial learning.

In Figure 4c we compare the importance of the spatial $||e_t^a - e_t^b||^2$ and temporal $||h_t^a - h_t^b||^2$ representations learned by VIRL. Using the recurrent representation (*temporal_lstm*) alone allows learning to progress quickly but can have difficulty informing the policy of how to best match the desired example. On the other hand, using only the encoding between single frames (*spatial_conv*) slows learning due to limited reward for out-of-phase behaviour. We achieved the best results by combining the representations from these two models. The assistance of spatial rewards is also seen in Figure 4b, where the *manual* reward learns the slowest.

**Ablation**    We conduct ablation studies in Figure 5a to compare the effects of data augmentation methods, network models and the use of additional data from other tasks. For the more complex *humanoid3d* control problems the data augmentation methods, including Early Episode Sequence Priority (EESP), increases average policy quality marginally. The use of mutlitask data Figure 8c and the additional representational losses Figure 8a greatly improve the methods ability to learn. More ablation results are available in the supplementary material.

**Sequence Encoding**    Using the learned sequence encoder a collection of motions from different classes are processed to create a TSNE embedding of the encodings (Maaten & Hinton, 2008). In Figure 5c we plot motions both generated from the learned policy $\pi$ and the expert trajectories

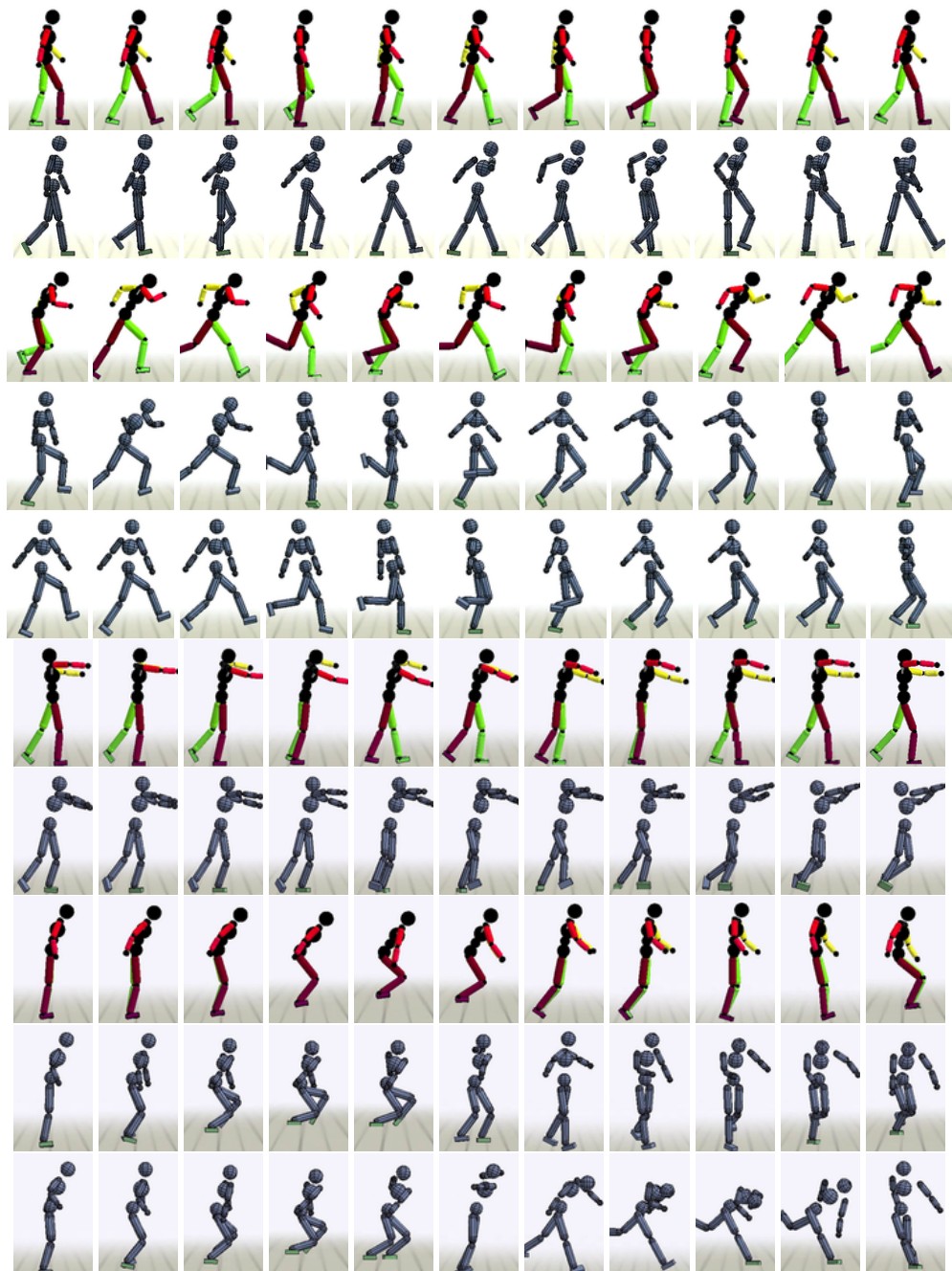

Figure 3: Rasterized frames of the agent's motion after training on humanoid3d walking (row 1,2), running (row 3-5), zombie (row 6,7) and jumping(row 8-10). The multi-coloured agent is a rendering of the imitation video. A video of these results is available here: https://youtu.be/s1KiIrV1YY4

$\pi_E$. Overlaps in specific areas of the space for similar classes across learned $\pi$ and expert $\pi_E$ data indicate a well-formed distance metric that does not sperate expert and agent examples. There is also a separation between motion classes in the data, and the cyclic nature of the walking cycle is visible.

In this section, we have described the process followed to create and analyze VIRL. Due to a combination of data augmentation techniques, VIRL can imitate given only a single demonstration. We have shown some of the first results to learn imitative policies from video data using a recurrent net-

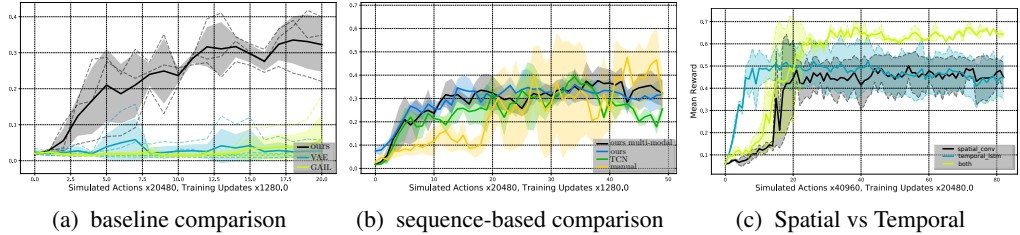

| (a) baseline comparison | (b) sequence-based comparison | (c) Spatial vs Temporal |
|---|---|---|

Figure 4: Baseline comparisons between our sequence-based method, GAIL and TCN (4a) on the humanoid2d environment. Two additional baseline comparison between VIRL and TCN in 4b. In 4c the benefit of combing spatial and temporal distances is shown. In these plots, the large solid lines are the average performance of a collection of policy training simulations. The dotted lines of the same colour are the specific performance values for each policy training run.

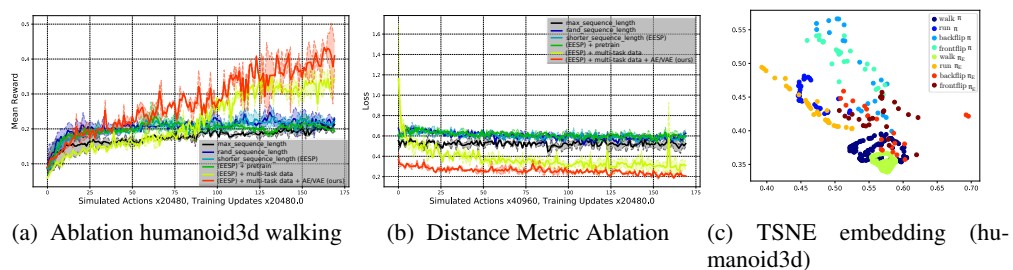

| (a) Ablation humanoid3d walking | (b) Distance Metric Ablation | (c) TSNE embedding (humanoid3d) |
|---|---|---|

Figure 5: Ablation analysis of VIRL. We find that training RL policies is sensitive to the size and distribution of rewards. A few modifications assist in the siamese network's ability to compute useful distances. Including VAE and AE losses to assist in representation learning. The addition of multi-task training data is also important for learning better policies.

work. Interestingly, the method displays new learning efficiencies that are important to the method success by separating the imitation problem into spatial and temporal aspects. For best results, we found that the inclusion of additional regularizing losses on the recurrent siamese network, along with some multi-task supervision, was key to producing results.

## 6 DISCUSSION AND CONCLUSION

In this work, we have created a new method for learning imitative policies from a single demonstration. The method uses a Siamese recurrent network to learn a distance function in both space and time. This distance function, trained on noisy partially observed video data, is used as a reward function for training an RL policy. Using data from other motion styles and regularization terms, VIRL produces policies that demonstrate similar behaviour to the demonstration.

Learning a distance metric is enigmatic, the distance metric can compute inaccurate distances in areas of the state space it has not yet seen. This inaccuracy could imply that when the agent explores and finds truly *new* and promising trajectories, the distance metric computes incorrect distances. We attempt to mitigate this effect by including training data from different tasks. We believe VIRL will benefit from a more extensive collection of multi-task data and increased variation of each task. Additionally, if the distance metric confidence is available, this information could be used to reduce variance and overconfidence during policy optimization.

It is probable learning a reward function while training adds additional variance to the policy gradient. This variance may indicate that the bias of off-policy methods could be preferred over the added variance of on-policy methods used here. We also find it important to have a small learning rate for the distance metric. The low learning rate reduces the reward variance between data collection phases and allows learning a more accurate value function. Another approach may be to use partially observable RL that can learn a better value function model given a changing RNN-based

reward function. Training the distance metric could benefit from additional regularization such as constraining the kl-divergence between updates to reduce variance. Learning a sequence-based policy as well, given that the rewards are now not dependent on a single state observation is another area for future research.

We compare our method to GAIL, but we found GAIL has limited temporal consistency. This method led to learning jerky and overactive policies. The use of a recurrent discriminator for GAIL may mitigate some of these issues and is left for future work. It is challenging to produce results better than the carefully manually crafted reward functions used by the RL simulation environments that include motion phase information in the observations (Peng et al., 2018a; 2017). However, we have shown that our method that can compute distances in space and time has faster initial learning. Potentially, a combination of starting with our method and following with a manually crafted reward function could lead to faster learning of high-quality policies. Still, as environments become increasingly more realistic and grow in complexity, we will need more robust methods to describe the desired behaviour we want from the agent.

Training the distance metric is a complicated balancing game. One might expect that the distance metric should be trained early and fast so that it quickly understands the difference between a good and bad demonstration. However, quickly learning confuses the agent, rewards can change, which cause the agent to diverge off toward an unrecoverable policy space. Slower is better, as the distance metric may not be accurate, it may be locally or relatively reasonable, which is enough to learn a good policy. As learning continues, these two optimizations can converge together.

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

# 7 APPENDIX

This section includes additional details related to VIRL.

## 7.1 IMITATION LEARNING

Imitation learning is the process of training a new policy to reproduce the behaviour of some expert policy. BC is a fundamental method for imitation learning. Given an expert policy $\pi_E$ possibly represented as a collection of trajectories $\tau < (s_0, a_0), \ldots, (s_T, a_T) >$ a new policy $\pi$ can be learned to match this trajectory using supervised learning.

$$\max_\theta \mathbb{E}_{\pi_E}[\sum_{t=0}^{T} \log \pi(a_t|s_t, \theta_\pi)] \tag{5}$$

While this simple method can work well, it often suffers from distribution mismatch issues leading to compounding errors as the learned policy deviates from the expert's behaviour.

## 7.2 INVERSE REINFORCEMENT LEARNING

Similar to BC, Inverse Reinforcement Learning (IRL) also learns to replicate some desired behaviour. However, IRL makes use of the RL environment without a defined reward function. Here we describe maximal entropy IRL (Ziebart et al., 2008). Given an expert trajectory $\tau < (s_0, a_0), \ldots, (s_T, a_T) >$ a policy $\pi$ can be trained to produce similar trajectories by discovering a distance metric between the expert trajectory and trajectories produced by the policy $\pi$.

$$\max_{c \in C} \min_\pi (\mathbb{E}_\pi[c(s, a)] - H(\pi)) - \mathbb{E}_{\pi_E}[c(s, a)] \tag{6}$$

where $c$ is some learned cost function and $H(\pi)$ is a causal entropy term. $\pi_E$ is the expert policy that is represented by a collection of trajectories. IRL is searching for a cost function $c$ that is low for the expert $\pi_E$ and high for other policies. Then, a policy can be optimized by maximizing the reward function $r_t = -c(s_t, a_t)$.

## 7.3 AUTO-ENCODER FRAMEWORK

**Variational Auto-encoders** Previous work shows that VAEs can learn a lower dimensional structured representation of a distribution (Kingma & Welling, 2014). A VAE consists of two parts an encoder $q_\phi$ and a decoder $p_\psi$. The encoder maps states to a latent encoding $z$ and in turn the decoder transforms $z$ back to states. The model parameters for both $\phi$ and $\psi$ are trained jointly to maximize

$$\mathcal{L}_{VAE}(\phi, \psi, s) = -\beta D_{KL}(q_\phi(z||s)||p(z) + \mathbb{E}_{q_\phi(z||s)}[\log p_\psi(s||z)] \tag{7}$$

, where $D_{KL}$ is the Kullback-Leibler divergence, $p(s)$ is some prior and $\beta$ is a hyper-parameter to balance the two terms. The encoder $q_\phi$ takes the form of a diagonal Gaussian distribution $q_\phi = \mathcal{N}(\mu_\phi(s), \sigma^2(s))$. In the case of images, the decoder $p_\psi$ parameterized a Bernoulli distribution over pixel values. This simple parameterization is akin to training the decoder with a cross entropy loss over normalized pixel values.

**Sequence Auto-encoding** The goal of sequence to sequence translation is to learn the conditional probability $p(y_0, \ldots, y_{T'}|x_0, \ldots, x_T)$, where $\mathbf{x} = x_0, \ldots, x_T$ and $\mathbf{y} = y_0, \ldots, y_{T'}$ are sequence Here we want to explicitly learn a latent variable $z_{RNN}$ that compresses the information in $x_0, \ldots, x_T$. An RNN can model this conditional probability by calculating $v = \prod_{t=0}^{T} p(y_T|\{x_0, \ldots, x_T\})$ of the sequence $\mathbf{x}$ that can, in turn, be used to condition the decoding of the sequence $\mathbf{y}$ (Rumelhart et al., 1985).

$$p(\mathbf{y}) = \prod_{t=0}^{T} p(y_T|\{y_0, \ldots, y_{T-1}\}, v) \tag{8}$$

, This method has been used for learning compressed representations for transfer learning (Zhu et al., 2016) and 3D shape retrieval (Zhuang et al., 2015).

### 7.4 DATA

The mocap used in the created environment come from the CMU mocap database and the SFU mocap database.

**Data Augmentation and Training**  We apply several data augmentation methods to produce additional data for training the distance metric. Using methods analogous to the cropping and warping methods popular in computer vision (He et al., 2015) we randomly *crop* sequences and randomly *warp* the demonstration timing. The *cropping* is performed by both initializing the agent to random poses from the demonstration motion and terminating episodes when the agent's head, hands or torso contact the ground. As the agent improves, the average length of each episode increases and so to will the average length of the cropped window. The motion *warping* is done by replaying the demonstration motion at different speeds. Two additional methods influence the data distribution. The first method is Reference State Initialization (RSI) (Peng et al., 2018a), where the initial state of the agent and expert is randomly selected from the expert demonstration. With this property, the environment can also be thought of as a form of memory replay. The environment allows the agent to go back to random points in the demonstration as if replaying a remembered demonstration. The second is EESP where the probability a sequence $\mathbf{x}$ is cropped starting at $i$ is $p(i) = \frac{len(\mathbf{x})-i}{\sum i}$, increasing the likelihood of starting earlier in the episode.

### 7.5 TRAINING DETAILS

The learning simulations are trained using Graphics Processing Unit (GPU)s. The simulation is not only simulating the interaction physics of the world but also rendering the simulation scene to capture video observations. On average, it takes 3 days to execute a single training simulation. The process of rendering and copying the images from the GPU is one of the most expensive operations with VIRL. We collect 2048 data samples between training rounds. The batch size for Trust Region Policy Optimization (TRPO) is 2048. The kl term is 0.5.

The simulation environment includes several different tasks that are represented by a collection of motion capture clips to imitate. These tasks come from the tasks created in the DeepMimic works (Peng et al., 2018a). We include all humanoid tasks in this dataset.

In Algorithm 1 we include an outline of the algorithm used for the method. The simulation environment produces three types of observations, $s_{t+1}$ the agent's proprioceptive pose, $s_{t+1}^v$ the image observation of the agent and $m_{t+1}$ the image-based oberservation of the expert demonstration. The images are $64 \times 64$.

### 7.6 DISTANCE FUNCTION TRAINING

Our Siamese training loss consists of

$$\mathcal{L}_{SN}(s_i, s_p, s_n) = y * ||f(s_i) - f(s_p)|| + ((1 - y) * (\max(\rho - (||f(s_i) - f(s_n)||), 0))), \quad (9)$$

where $y = 1$ is a positive example $s_p$, pair where the distance should be minimal and $y = 0$ is a negative example $s_n$, pair where the distance should be maximal. The *margin* $\rho$ is used as an attractor or anchor to pull the negative example output away from $s_i$ and push values towards a 0 to 1 range. $f(\cdot)$ computes the output from the underlying network. The distance between two states is calculated as $d(s, s') = ||f(s) - f(s')||$ and the reward as $r(s, s') = -d(s, s')$. Data used to train the Siamese network is a combination of trajectories $\tau = \langle s_0, \ldots, s_T \rangle$ generated from simulating the agent in the environment and the expert demonstration. For our recurrent model the same loss is used; however, the states $s_p, s_n, s_i$ are sequences. During RL training we compute a distance given the sequence of states observed so far in the episode. This method allows us to train a distance function in state space where all we need to provide is labels that denote if two states, or sequences, are similar or not.

In Figure 6b we show the training curve for the recurrent siamese network. The model learns smoothly, considering that the training data used is continually changing as the RL agent explores. In Figure 6a the learning curve for the siamese RNN is shown after performing pretraining. We can see the overfitting portion the occurs during RL training. This overfitting can lead to poor reward prediction during the early phase of training.

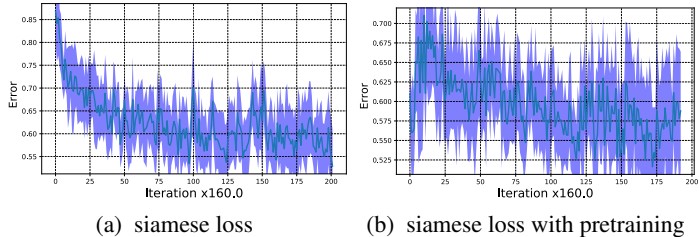

(a) siamese loss      (b) siamese loss with pretraining

Figure 6: Training losses for the siamese distance metric. Higher is better as it indicates the distance between sequences from the same class are closer.

It can be challenging to train a sequenced based distance function. One particular challenge is training the distance function to be accurate across the space of possible states. We found a good strategy was to focus on the beginning of episode data. When the model is not accurate on states it saw earlier in the episode; it may never learn how to get into good states later that the distance function understands better. Therefore, when constructing batches to train the RNN on, we give a higher probability of starting earlier in episodes. We also give a higher probability to shorter sequences. As the agent gets better average episodes length increase, so to will the randomly selected sequence windows.

## 7.7 DISTANCE FUNCTION USE

We find it helpful to *normalize* the distance metric outputs using $r = exp(r^2 * w_d)$ where $w_d = -5.0$ scales the filtering width. Early in training the distance metric often produces large, noisy values. Also, the RL method regularly updates reward scaling statistics; the initial high variance data reduces the significance of better distance metric values produced later on by scaling them to small numbers. The improvement of using this normalize reward is shown in Figure 7a.

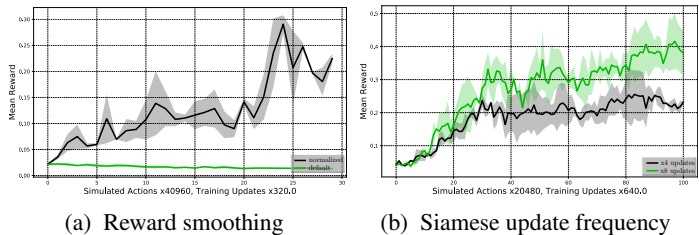

(a) Reward smoothing      (b) Siamese update frequency

Figure 7: Ablation analysis of VIRL. We find that training RL policies is sensitive to the size and distribution of rewards. The siamese network benefits from several training adjustments that make it more suitable for RL.

## 8 POSITIVE AND NEGATIVE EXAMPLES

We use two methods to generate positive and negative examples. The first method is similar to TCN, where we can assume that sequences that overlap more in time are more similar. For each episode two sequences are generated, one for the agent and one for the imitation motion. Here we list the methods used to alter sequences for positive pairs.

1. Adding Gaussian noise to each state in the sequence (mean $= 0$ and variance $= 0.02$)
2. Out of sync versions where the first state is removed from the first sequence and the last state from the second sequence
3. Duplicating the first state in either sequence
4. Duplicating the last state in either sequence

We alter sequences for negative pairs by

1. Reversing the ordering of the second sequence in the pair.

2. Randomly picking a state out of the second sequence and replicating it to be as long as the first sequence.

3. Randomly shuffling one sequence.

4. Randomly shuffling both sequences.

The second method we use to create positive and negative examples is by including data for additional classes of motion. These classes denote different task types. For the humanoid3d environment, we generate data for walking-dynamic-speed, running, backflipping and frontflipping. Pairs from the same tasks are labelled as positive, and pairs from different classes are negative.

## 8.1 Additional Ablation Analysis

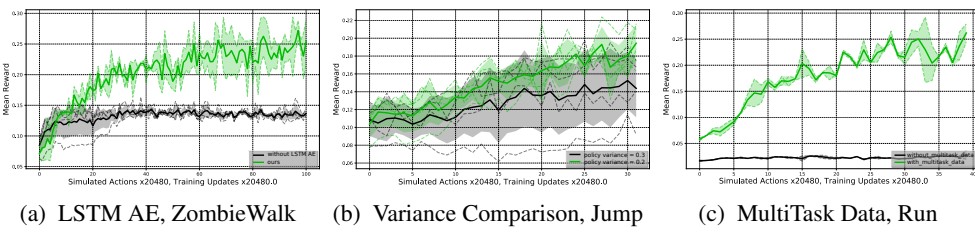

(a) LSTM AE, ZombieWalk    (b) Variance Comparison, Jump    (c) MultiTask Data, Run

Figure 8: Additional ablation analysis of VIRL on humanoid3d.

## 8.2 RL Algorithm Analysis

It is not clear which RL algorithm may work best for this type of imitation problem. A number of RL algorithms were evaluated on the humanoid2d environment Figure 9a. Surprisingly, TRPO (Schulman et al., 2015) did not work well in this framework, considering it has a controlled policy gradient step, we thought it would reduce the overall variance. We found that Deep Deterministic Policy Gradient (DDPG) (Lillicrap et al., 2015) worked rather well. This result could be related to having a changing reward function, in that if the changing rewards are considered off-policy data, it can be easier to learn. This can be seen in Figure 9b where DDPG is best at estimating the future discounted rewards in the environment. We also tried Continuous Actor Critic Learning Automaton (CACLA) (Van Hasselt, 2012) and Proximal Policy Optimization (PPO) (Schulman et al., 2017); we found that PPO did not work particularly well on this task; this could also be related to added variance.

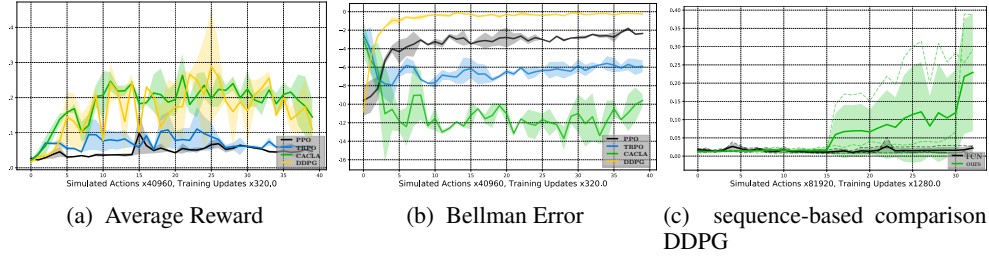

(a) Average Reward    (b) Bellman Error    (c) sequence-based comparison DDPG

Figure 9: RL algorithm comparison on humanoid2d environment.

## 8.3 Additional Imitation Results

Our first experiments evaluate the methods ability to learn a complex cyclic motion for a simulated humanoid robot given a single motion demonstration, similar to (Peng & van de Panne, 2017), but using video instead. The agent is able to learn a robust walking gate even though it is only given noisy partial observations of a demonstration Figure 10.

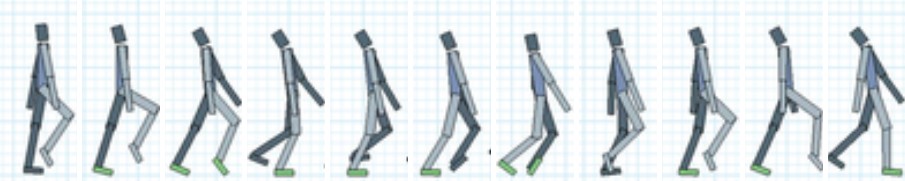

Figure 10: Still frame shots from a policy trained in the humanoid2d environment.

