# OpenReview forum: "Visual Imitation with Reinforcement Learning using Recurrent Siamese Networks"
_ICLR.cc/2020/Conference — Reject_

### Official Review · AnonReviewer2 · 2019-10-23
**Official Blind Review #2**

**Rating:** 8

**Review:**

This paper presents visual imitation with reinforcement learning (VIRL), an algorithm for learning to imitate expert trajectories based solely on visual observations, and without access to the expert’s actions.  The algorithm is similar in form to GAIL and its extensions, learning a reward function which captures the similarity between an observed behavior and the expert's demonstrations, while simultaneously using reinforcement learning to find a policy maximizing this reward, such that the learned policy will replicate the demonstrated behavior as well as possible.  A key feature of this method is that the learned reward function is defined by a learned distance metric, which evaluates the similarity between the agent's current trajectory, and the nearest demonstrated expert trajectory.

The network describing the distance metric is recurrent, such that the distance is defined between trajectories rather than individual states.  The distance function network is trained via a negative sampling approach, where expert trajectories are randomly reordered to produce examples that dissimilar to the expert trajectories.   The distance network also defines a variational autoencoder, and the reconstruction of the target trajectories is treated as an auxiliary task to help train better representations of the trajectory space.

While previous work has considered the problem of visual imitation learning, the approach taken here is novel in its architecture and loss function, and significantly outperforms the baselines in terms of the similarity between the resulting behavior and the expert behavior.

The clarity of the technical presentation could be improved, however.  In particular, it would be helpful for the reader if the definitions of the negative sampling loss and the autoencoder losses were given before the combined loss, and if we saw the form of the loss for both positive and negative sequence pairs.  Equation 4 could also be made explicit, with the full summation term included.

**Experience Assessment:**

I have read many papers in this area.

**Review Assessment: Checking Correctness Of Derivations And Theory:**

I assessed the sensibility of the derivations and theory.

**Review Assessment: Checking Correctness Of Experiments:**

I assessed the sensibility of the experiments.

**Review Assessment: Thoroughness In Paper Reading:**

I read the paper at least twice and used my best judgement in assessing the paper.

---

> ### Author Response · Authors · 2019-11-07
> **Thank you, will will make the writing more clear.**
>
> Thank you for the feedback on the paper. There is a large amount of content necessary to fit into the paper. In the next version of the paper, we will move the negative sampling and auto-encoder losses before the combined loss. We will also update equation 4.

---

> > ### Comment · AnonReviewer2 · 2019-11-14
> > **Visually rich environments**
> >
> > Thank you for taking the time to address these comments.
> >
> > As a final note, I believe one of the main concerns with this work is that the experimental domain (humanoid), while a challenging control problem, is perhaps not as visually challenging as other RL domains.  As the primary motivation for this work is to enable imitation learning from visual observations, it would strengthen the argument for the value of the proposed approach (relative to existing methods), if it were evaluated in domains with more complex visual representations.  A good example might be the CoinRun domain: https://github.com/openai/coinrun, which specifically tests the robustness of RL methods to different visual representations.

---

### Official Review · AnonReviewer3 · 2019-10-25
**Official Blind Review #3**

**Rating:** 3

**Review:**

This paper presents an imitation learning method that deploys previously well-studied techniques such as siamese networks, inverse RL, learning distance functions for IRL and tracking.

+ the paper studies an important problem of IL using visual data.
+ I found the ablation studies in the appendix quite useful in understanding the efficiency of the proposed method.

-In terms of novelty, the proposed approach is a combination of several past works so the technical novelty is limited. Additionally, it is not clear how impactful the proposed method can be given that it is only tested on a synthetic domain which is the same as the train domain. So, from the current experimental results it is not clear if this approach would be effective to be applied in a real system (e.g. robots) on the practical side.

-There are not enough evaluation done to compare with the most updated state-of-the-art baselines. The evaluations are done on just a single synthetic domain with a single character. Therefore, the train and test videos are very similar.



**Experience Assessment:**

I have published one or two papers in this area.

**Review Assessment: Checking Correctness Of Derivations And Theory:**

I assessed the sensibility of the derivations and theory.

**Review Assessment: Checking Correctness Of Experiments:**

I assessed the sensibility of the experiments.

**Review Assessment: Thoroughness In Paper Reading:**

I read the paper at least twice and used my best judgement in assessing the paper.

---

> ### Author Response · Authors · 2019-11-07
> **Discussion of novelty**
>
> We appreciate the comments on the work.
>
> The main concern raised in this review is the novelty of the work. While Siamese networks and learning to imitate from vision are not novel by themselves, there has not been work that combines these methods, especially with sequence-based methods. The reviewer points out we should “compare with the most updated state-of-the-art baselines”. We examine a setting for which few state-of-the-art baselines exist and have only been applied to simpler tasks. We do compare to recent methods, TCN and GAIL, we find that our method outperforms these methods (Figure 4(a)). Also, while often training RL using image data slows down learning, we show in Figure 4(b,c) that our combination of spatial and temporal distances leads to a rich reward landscape that results in faster learning. While we do not apply our system directly on a robot, we do show that our method outperforms TCN, which was explicitly used on robots. The evaluations performed in the paper are over two separate simulated domains, shown in figures 3 and 10, and for one of these domains, we apply the method over multiple imitation tasks.
>
> Additionally, the simulated environment used in this work was created and is the only simulation library available that provides visual data for expert demonstrations across multiple-tasks. We plan to add more tasks and release this simulation library to enable others to explore the problem of visual imitation learning further. We leave applying this method to a real robot as interesting future work.

---

> ### Author Response · Authors · 2019-11-13
> **Rebuttal Discussion**
>
> Dear Reviewer,
>
> Could you let us know if our response has addressed the concerns raised in your review? We would be happy to provide further revisions or experiments to address any remaining issues and would appreciate a response from you on the points that we raised.

---

### Official Review · AnonReviewer1 · 2019-10-26
**Official Blind Review #1**

**Rating:** 3

**Review:**

The idea of the paper is to learn a distance function between observed and the agent’s behaviors. Once they have the distance function, they can learn the agent’s policy efficiently given a single demonstration of each task. In their formulation, the distance function and the policy are jointly learned.

The idea is reasonable and the performance outperforms baselines like GAIL and VAE. However, the paper is not-well written with many relevant equations defined in the supplementary material. The unsupervised data labeling part seems Adhoc with many details in the supplementary material. I wonder if the process stable or not. How many lower than the average performance of the proposed method as shown in F.g 4 are caused by unsupervised data labeling?

In Fig. 4b, the manual performance is very strong once converged. Although the proposed method initially reaches high reward, after twice many iterations the manual performance even outperforms the proposed method on average many times. Hence, I am not very convinced about the proposed method will be the best-picked method in practice.

Overall, I think the idea is good. But the paper is poorly written and I concern the most about the stability of the unsupervised data labeling process. The experimental results are also not super convincing. Hence, I recommend for weak rejection.

**Experience Assessment:**

I have published one or two papers in this area.

**Review Assessment: Checking Correctness Of Derivations And Theory:**

I assessed the sensibility of the derivations and theory.

**Review Assessment: Checking Correctness Of Experiments:**

I assessed the sensibility of the experiments.

**Review Assessment: Thoroughness In Paper Reading:**

I made a quick assessment of this paper.

---

> ### Author Response · Authors · 2019-11-07
> **Oracle/manual rewards are not available in the real world, our contribution is to learn a reward function that can.**
>
> Thank you for the comments on the work, we regret the confusion.
>
> The main concern raised in the review is that the manual reward function results in higher performance than VIRL. The manual result in Figure 4(b) is meant to be illustrative of the performance of VIRL compared to a manual or oracle reward function. In this “manual” result, the learning system has access to the noise-free true state of the agent and expert to compute rewards. However, we point out that the contribution of the work is a method that can learn how to imitate quickly when the agent, similar to a human, only has access to noisy images of the expert demonstration and agent. Figure 4(b) shows that our method that receives noisy image data of the demonstration can accelerate learning compared to the manual/oracle reward function that is not available in real-world settings.
>
> The second concern in this review is the unsupervised data labelling process. This data labelling scheme is a natural extension of the process used in the “Time Contrastive Networks” and Dwibedi et al. (2018) papers with a few novel additions as a result of using sequences instead of states. We tried many different labelling schemes for this work. Some of these schemes introduced forms of bias, and we found that the minimal set described in the paper gave us the best results across all tasks we trained. We did not observe instability related to the data relabeling process. However, we will add further details about the labelling process and move this to the main part of the paper. To make the writing more precise, we will also move the equations from the appendix to earlier in the paper.
>
> Learning how to imitate from a single video demonstration of an articulated 3D humanoid is challenging and has not been demonstrated before. In this work, we show that we can train policies to imitate many different motions. In addition, we also show that using the method described in the paper, which uses a combination of spatial and temporal distance, increases learning speed.

---

> ### Author Response · Authors · 2019-11-13
> **Rebuttal Discussion**
>
> Dear Reviewer,
>
> Could you let us know if our response has addressed the concerns raised in your review? We would be happy to provide further revisions or experiments to address any remaining issues and would appreciate a response from you on the points that we raised.

---

### Decision · Program_Chairs · 2019-12-19

**Decision:**

Reject

**Comment:**

The main concern raised by reviewers is limited novelty, poor presentation, and limited experiments. All the reviewers appreciate the difficulty and importance of the problem. The rebuttal helped clarify novelty, but the other concerns remain.